# Elevated Expression of miR-200c/141 in MDA-MB-231 Cells Suppresses *MXRA8* Levels and Impairs Breast Cancer Growth and Metastasis In Vivo

**DOI:** 10.3390/genes13040691

**Published:** 2022-04-14

**Authors:** Kaitlyn E. Simpson, Katrina L. Watson, Roger A. Moorehead

**Affiliations:** Department of Biomedical Sciences, Ontario Veterinary College, University of Guelph, Guelph, ON N1G 2W1, Canada; ksimps07@uoguelph.ca (K.E.S.); kawatson@uoguelph.ca (K.L.W.)

**Keywords:** miR-200, claudin-low breast cancer, tumor growth, metastasis, *MXRA8*, MDA-MB-231

## Abstract

Breast cancer cells with mesenchymal characteristics, particularly the claudin-low subtype, express extremely low levels of miR-200s. Therefore, this study examined the functional impact of restoring miR-200 expression in a human claudin-low breast cancer cell line MDA-MB-231. MDA-MB-231 cells were stably transfected with a control vector (MDA-231EV) or the miR-200c/141 cluster (MDA-231c141). Injection of MDA-231c141 cells into the 4th mammary gland of NCG mice produced tumors that developed significantly slower than tumors produced by MDA-231EV cells. Spontaneous metastasis to the lungs was also significantly reduced in MDA-231c141 cells compared to MDA-231EV cells. RNA sequencing of MDA-231EV and MDA-231c141 tumors identified genes including *MXRA8* as being downregulated in the MDA-231c141 tumors. *MXRA8* was further investigated as elevated levels of *MXRA8* were associated with reduced distant metastasis free survival in breast cancer patients. Quantitative RT-PCR and Western blotting confirmed that *MXRA8* expression was significantly higher in mammary tumors induced by MDA-231EV cells compared to those induced by MDA-231c141 cells. In addition, MXRA8 protein was present at high levels in metastatic tumor cells found in the lungs. This is the first study to implicate *MXRA8* in human breast cancer, and our data suggests that miR-200s inhibit growth and metastasis of claudin-low mammary tumor cells in vivo through downregulating *MXRA8* expression.

## 1. Introduction

MicroRNAs (miRNAs) are small, non-coding RNAs that regulate mRNA expression by binding to mRNAs and in turn, prevent mRNA translation [1,2]. The interaction between miRNAs and mRNAs is primarily regulated by a region known as the seed sequence that binds to complementary sequences in the 3′ untranslated regions of mRNAs [3,4,5,6,7,8,9,10,11].

MicroRNAs have been implicated in regulating a number of cancers, including breast cancer. One family of miRNAs implicating in breast cancer is the miR-200 family. This family consists of five members organized into two clusters, the miR-200b/200a/429 cluster and the miR-200c/141 cluster [12,13,14]. These five miRNAs have highly similar seed sequences and the miR-200b, miR-200c, and miR-429 share a common seed sequence (AAUACUG), while miR-200a and miR-141 share the same seed sequence (AACACUG), which differs from the seed sequence of miR-200b, miR-200c and miR-429 by one nucleotide [15]. The most completely characterized function of the miR-200 family is their role in maintaining epithelial identity; miR-200 family members reduce the expression of mesenchymal transcription factors, such as *ZEB1/2*, *TWIST1/2*, *SNAI1/2* [14,16,17,18], and increase the expression of epithelial genes such as E-cadherin [19,20]. 

Given their roles in epithelial identity, it is not surprising that miR-200s are expressed in human luminal breast cancer but are only expressed at very low levels in cancers with mesenchymal characteristics, such as claudin-low breast cancer [21,22,23,24,25]. Claudin-low breast cancers share features with triple negative breast cancer (TNBC), including low levels of the estrogen receptor (ER), progesterone receptor (PR) and human epidermal growth factor receptor 2 (HER2) [26]. Claudin-low breast cancers are differentiated from other TNBCs by the low levels of cell-cell adhesion and tight junction genes such as claudins 3, 4, 7 [26,27,28].

Our previous work investigating the function of miR-200s in human claudin-low breast cancer cell lines found that re-expression of the miR-200c/141 cluster in human MDA-MB-231 cells reverted these cells to a more epithelial morphology, significantly reduced proliferation, and impaired migration and invasion in vitro [29]. Thus, the goal of this study was to evaluate the impact of miR-200c/141 overexpression on in vivo mammary tumor growth and metastasis. In the current study, it was shown that re-expression of the miR-200c/141 cluster in MDA-MB-231 cells significantly impaired mammary tumor growth in vivo. Re-expression of the miR-200c/141 cluster in MDA-MB-231 cells was also capable of significantly inhibiting spontaneous metastasis to the lungs. The reduced levels of *MXRA8* in the mammary tumors induced by MDA-231c141 cells implicate this gene in regulating mammary tumor growth and metastasis.

## 2. Materials and Methods

### 2.1. Cell Lines and Culture Conditions

MDA-231EV cells were created by infecting MDA-MB-231 cells with copGFP lentiviral particles (cat sc-108,084, Santa Cruz Biotechnology Inc., Dallas, TX, USA), and stable cells were selected by continual culture in 5 µg/mL puromycin (InvivoGen, San Diego, CA, USA). MDA-231c141 cells were created by transfecting MDA-MB-231 cells with pLenti 4.1 Ex miR200c-141 (cat #35,534, Addgene, Watertown, MA, USA) and stable cells were selected by continual culture in 5 µg/mL puromycin (InvivoGen, San Diego, CA, USA). The in vitro characterization of MDA-231EV and MDA-231c141 cells have previously been described [29].

### 2.2. Animals and Ethics

Mice were housed and cared for following guidelines established by the Central Animal Facility at the University of Guelph, and the guidelines established by the Canadian Council of Animal Care. This study was approved by the Animal Care Committee at the University of Guelph (AUP #3995).

MDA-231EV and MDA-231c141 cells were collected from logarithmically growing cultures, and 1 × 10^6^ cells were injected into the mammary glands of female NCG (NOD-*Prkdc^em26Cd52^Il2rg^em26Cd22^*/NjuCrl) mice (Charles River, Wilmington MA, USA). Ten mice were injected with each cell line and injections occurred at three different times, separated by several weeks. For each independent round of injections, both MDA-231EV and MDA-231c141 cells were injected. All mice were monitored twice per week by palpating the mammary glands. Once a palpable mammary tumor was identified, tumor size was measured using digital calipers. MDA-231EV and MDA-231c141 tumor bearing mice were collected once the MDA-231EV mammary tumors reached ~10% of the mouse’s body weight (maximum size allowed by the Canadian Council on Animal Care).

### 2.3. Tumor Specific Growth Rates (SGR)

Tumor specific growth rates were calculated using the formula SGR = ln (V_2_/V_1_)/(t_2_ − t_1_), where V_1_ and V_2_ are the tumor volumes at the time of palpation (t_1_) and euthanasia (t_2_), respectively.

### 2.4. Histology and Immunohistochemistry

Mammary tumors and major organs were fixed in 10% formalin overnight and embedded in paraffin. Sections were cut and stained with hematoxylin and eosin for histologic analysis. Immunohistochemistry was performed as previously described [30], using a 1:200 dilution of an anti-vimentin antibody (ab16700, Abcam, Toronto, ON, Canada) or a 1:100 dilution of an anti-MXRA8 antibody (ab185444, Abcam). Tissue sections were scanned using a Motic Easyscan digital slide scanner (Motic, Richmond, BC, Canada). 

Lung tumor burden was calculated using lung tissue sections from MDA-231EV and MDA-231c141 tumor bearing mice stained with the human anti-vimentin antibody. Comparison of the H&E stained sections with vimentin stained sections confirmed that metastatic cells stained positive for vimentin. Vimentin stained sections were analyzed using ImageJ software [31], and the area of the section occupied by vimentin positive cells was determined using the color threshold function of ImageJ software [31]. Lung area was determined using the freehand drawing tool to outline each piece of lung tissue and total lung area on each section was calculated using Image J software [31].

### 2.5. RNA Extraction and RNA Sequencing

RNA was isolated from four independent MDA-231EV and four independent MDA-231c141 tumors. All samples had a 260/280 ratio > 1.95 (Nanodrop) and ~2 μg of RNA was sent to Genome Quebec. Genome Quebec analyzed the RNA on an Agilent Bioanalyzer 2100 (Agilent, Santa Clara, CA, USA), and the samples had an RIN of 9.5 or greater. Libraries were generated from 250 ng of total RNA which included mRNA enrichment using the NEBNext Poly(A) Magnetic Isolation Module (New England BioLabs, Whitby, ON, Canada), cDNA synthesis using NEBNext RNA First Strand Synthesis and NEBNext Ultra Directional RNA Second Strand Synthesis Modules (New England BioLabs, Whitby, ON, Canada). The libraries were normalized, pooled and denatured in 0.05N NaOH and neutralized using HT1 buffer. The pool was loaded at 225 pM on an Illumina NovaSeq S4 lane using Xp protocol as per the manufacturer’s recommendations. The run was performed for 2 × 100 cycles (paired-end mode). A phiX library was used as a control and mixed with libraries at 1% level. Base calling was performed with RTA v3. Program bcl2fastq2 v2.20 was then used to demultiplex samples and generate fastq reads.

Fastq reads were analyzed using Genialis software (Genialis Inc., Houston, TX, USA) following the software’s standard RNA-seq pipeline, as previously described [29]. Hierarchical clustering was performed using Genialis software (Genialis Inc., Houston, TX, USA) with all genes and Pearson distance measure and average linkage clustering. Pathway analysis was performed using Enrichr software [32,33]. The RNA sequencing data has been uploaded to the GEO database as accession number GSE193479.

### 2.6. Real-Time PCR

For TaqMan real-time PCR, the same RNA that was extracted for sequencing was used. 100 ng of RNA was reversed transcribed using the TaqMan microRNA reverse transcription kit (Thermo Fisher Scientific, Burlington, ON, Canada) following the manufacturer’s protocol. Primers for reverse transcribing and amplifying cDNA for miR-141 (ID 000463), miR-200a (ID 000502), miR-200b (ID 001800), miR-200c (ID 002300), miR-429, RNU44 (ID 001094) and RNU48 (ID 001006) were obtained from Thermo Fisher Scientific (Burlington, ON, Canada). qPCR was performed using a CFX96 real-time PCR machine (Bio-Rad Laboratories, Mississauga, ON, Canada) using 2 μL of each cDNA reaction, 10 μL of TaqMan universal master mix II no UNG (Thermo Fisher Scientific, Burlington, ON, Canada), 1 μL of 20x primer, and 7 μL of water, under the following incubation conditions; 50 °C for 2 min, 95 °C for 10min, and then 40 cycles of 95 °C for 15 s and 60 °C for 1 min. miRNA levels were calculated using Bio-Rad CFX Manager 3.1 (Bio-Rad Laboratories, Mississauga, ON, Canada) and were normalized to the levels of RNU44 and RNU48.

For gene expression, 1 ug of total RNA (same RNA as used for sequencing) was reverse transcribed using 4 μL of qScript cDNA supermix (Quanta Biosciences, Beverly, MA, USA) per 20 μL reaction and the following incubation conditions on a thermal cycler; 25 °C for 5 min, 42 °C for 30 min, 85 °C for 5 min, and then held at 4 °C until used for qPCR. For qPCR, 2 μL of cDNA was mixed with 10 μL of SensiFast SYBR No-Rox (FroggaBio Inc, Concord, ON) and 7 μL of water and 1 µL of *MXRA8* primer (qHsaCED0045992) or *HPRT* primer (qHsaCID0016375). Both primers were obtained from Bio-Rad Laboratories (Mississauga, ON, Canada). Gene expression was quantified using a CFX96 real-time PCR machine (Bio-Rad Laboratories, Mississauga, ON, Canada) and the following program; 95 °C for 2 min and then 40 cycles of 95 °C for 5 s and 60 °C for 30 s. *HPRT* was used as the housekeeping gene and gene expression was quantified using CFX Manager software (Bio-Rad Laboratories, Mississauga, ON, Canada).

### 2.7. Western Blotting

Western blotting was performed as previously described [34]. Briefly, protein was isolated from four independent MDA-231EV and MDA-231c141 tumors. Forty micrograms of protein for each sample were separated on an 8% polyacrylamide and transferred to nitrocellulose membrane. Membranes were incubated with a 1:500 dilution of an anti-MXRA8 antibody (ab185444; Abcam, Toronto, ON, Canada) overnight at 4 °C. Proteins were detected with HRP-linked anti-rabbit IgG secondary antibody (1:2000) (New England Biolabs, Ltd., Whitby, ON, Canada) and Clarity Western ECL substrate (Bio-Rad Canada, Mississauga, ON, Canada). The protein was then imaged using the ChemiDocTMXRS+ System (Bio-Rad Canada, Mississauga, ON, Canada), and quantification was performed using Image Lab software (Bio-Rad Canada, Mississauga, ON, Canada). Following detection of MXRA8, the membranes were blocked and incubated with a 1:5000 dilution of an anti-β-actin antibody (cat. no. 8457; New England Biolabs, Whitby, ON, Canada). 

### 2.8. Kaplan–Meier Plotter

The online survival tool, Kaplan–Meier Plotter, was used to determine whether gene expression was associated with distant metastases-free survival (DMFS) [35]. This analysis was restricted to basal-like breast cancers as MDA-MB-231 cells represent a class of basal-like breast cancer. Patients were split by the median value [35].

### 2.9. Statistics

For analysis comparing the means of two different groups, a Student’s *t*-test was performed. Means were considered statistically different when *p* < 0.05.

## 3. Results

### 3.1. miR-200s Impair In Vivo Tumor Growth and Metastasis

Previously published work from our lab characterized the MDA-231EV and MDA-231c141 cells in vitro [29]. MDA-231EV cells were transfected with a control vector, while MDA-231c141 cells were transfected with a vector containing the miR-200c/141 cluster. To gain a better understanding of the function of miR-200s in tumorigenesis, MDA-231EV (1 × 10^6^ cells) or MDA-231c141 (1 × 10^6^ cells) were injected into the mammary fat pads of NCG (NOD-*Prkdc^em26Cd52^Il2rg^em26Cd22^*/NjuCrl) mice. As shown in Figure 1A, MDA-231c141 cells displayed reduced tumor growth compared to MDA-231EV cells (Figure 1A). Specific growth rates were calculated for each tumor, and the specific growth rate for MDA-231c141 tumors (1.48 ± 0.13) was significantly lower than MDA-231EV (3.24 ± 0.05) tumors.

Histological examination of the mammary tumors produced from intramammary injections of MDA-231EV or MDA-231c141 cells revealed that both tumors presented as solid sheets of tumor cells with variable amounts of stroma and necrosis (Figure 1B,C). 

To assess whether miR-200 expression observed in the cell lines were maintained in vivo, Taqman RT-PCR was performed. As shown in Figure 1D–H, tumors produced by the injection of MDA-231c141 cells had significantly higher levels of miR-200c (Figure 1D) and miR-141 (Figure 1E) than MDA-231EV tumors. 

MDA-231EV tumors spontaneously metastasized to the lungs in 100% of the mice (Figure 2A–C, Table 1), while MDA-231c141 tumors only metastasized to the lungs in 70% of the mice (Figure 2D–F, Table 1). Average tumor burden (based on positive staining for human vimentin) in the lungs of mice harboring MDA-231EV tumors were significantly higher than the average tumor burden of the lungs of mice harboring MDA-231c141 tumors (Figure 2G). 

MDA-231EV tumors also frequently metastasized to the pancreas and spleen (Table 1, Appendix A) and less frequently to the liver (Table 1, Appendix A). The metastatic lesions that developed near the pancreas and spleen were unusual in that they appeared to grow just outside these tissues with sporadic regions of tumor cell invasion (Appendix A).

### 3.2. Elevated MXRA8 Is Associated with Tumor Growth, Metastasis, and Patient Prognosis

RNA sequencing was performed on tumors derived from MDA-231EV and MDA-231c141 cells. Hierarchical clustering of the tumors revealed that the MDA-231EV and MDA-231c141 tumors segregated into discrete clusters (Appendix A). Using cut-offs of logFC ≥ 1 and FDR < 0.00001 it was determined that MDA-231c141 tumors had 1273 transcripts that were upregulated, and 1570 transcripts were downregulated compared to MDA-231EV tumors. Pathway analysis of the 2843 differentially expressed transcripts in the MDA-231c141 vs. MDA-231EV tumors using Enrichr software revealed the top ENCODE and ChEA Consensus TFS from ChIP-X was SUZ12, while the top MSigDB Hallmark pathway was epithelial mesenchymal transition, and the top GO Biological Process was axon guidance (Figure 3).

The top five differentially expressed transcripts in the MDA-231c141 vs. MDA-231EV comparison (based on FDR) are presented in Table 2, and all five were significantly downregulated in the MDA-231c141 tumors. Since all five transcripts were downregulated in the MDA-231c141 tumors, three databases (miRDB, microT-CDS, and miRWalk) were investigated to determine whether these transcripts contained potential miR-200c or miR-141 binding sites. The database miRDB uses the target prediction tool MirTarget [36,37] and microT-CDS identifies miRNA recognition elements in the 3′UTR and coding sequences [38], while miRWalk searches for potential binding sites in 3′-UTR, 5′-UTR and coding regions using TarPmiR [39,40]. As shown in Appendix A, only miRwalk but not miRDB or MicroT-CDS indicated that these five transcripts had potential miR-200c or miR-141 binding sites. 

To evaluate whether *CPE*, *GNG2*, *MXRA8*, *SELENBP1* or *AGR2* were associated with survival in breast cancer patients, these genes were examined using kmplot.com. Tumor types were restricted to the basal-like breast cancer subtype as the basal-like subtype is the closest subtype to claudin-low (MDA-MB-231 cells have characteristics of claudin-low breast cancer [26], and patients were split by the median expression values. Only elevated *MXRA8* (Figure 4) was significantly associated with a decrease in distant metastasis free survival in basal-like breast cancer patients.

The levels of *MXRA8* were then evaluated in the primary tumors using qRT-PCR (Figure 5A) and Western blotting (Figure 5B,C). At both the mRNA and protein level, *MXRA8* was significantly higher in the MDA-231EV tumors compared to the MDA-231c141 tumors. To evaluate the levels of MXRA8 protein in lung metastasis, immunohistochemistry was performed. The metastatic tumor cells expressed high levels of MXRA8 compared to the adjacent normal lung tissue (Figure 6A,B).

## 4. Discussion

Although it is generally accepted that miR-200s negatively regulate breast cancer growth and metastasis, studies directly assessing miR-200c/miR-141 function in vivo are limited, and the findings are not always consistent. Only four studies have investigated the impact of miR-200c overexpression on in vivo growth of human breast cancer cell lines. All four studies found that elevated levels of miR-200s inhibited primary mammary tumor growth following intramammary [41,42,43] or subcutaneous injection [44]. Similarly, four of the five studies investigating miR-200c or miR-200c/141 overexpression in murine mammary tumor cells found that elevated miR-200s significantly reduce primary mammary tumor growth [44,45,46,47]. One study found that elevated expression of miR-200c and miR-141 in the murine mammary tumor cell line, 4TO7, did not significantly affect tumor growth [48]. The majority of these studies are consistent with our findings and suggest that miR-200s negatively regulate growth of primary mammary tumors.

Only one other study has evaluated the impact of miR-200c overexpression on spontaneous metastasis of human breast cancer cells and, like our study, showed that elevated miR-200c could significantly reduce spontaneous metastasis to the lungs [49]. Interestingly, two studies evaluating lung metastasis of MDA-MB-231 cells overexpressing miR-200c or miR-200c/141 following tail vein injection found that elevated miR-200 levels were associated with an increase in lung metastases [50,51]. There have also been two studies on metastasis of murine mammary tumor cells, with one study using an experimental metastasis assay [47], and the other evaluating spontaneous metastasis from a primary mammary tumor [46]. Both studies found that miR-200c/141 expression was negatively associated with lung metastases [46,47]. Since spontaneous metastasis assays more accurately reflect metastasis in the patient, our study and the studies by Liu et al. [46] and Li et al. [49] suggest that elevated miR-200c/141 expression will reduce metastatic dissemination.

Transcriptome profiling of MDA-231EV and MDA-231c141 tumors revealed a number of genes regulated by the miR-200c/141 cluster that could influence mammary tumor growth and metastasis including *MXRA8*. *MXRA8* or matrix remodeling associated 8 is a transmembrane protein that can influence integrin signaling and can regulate cell–cell interactions. *MXRA8* also serves as the receptor for arthritogenic alphaviruses [52]. While *MXRA8* has been associated with esophageal, kidney, gingivobuccal cancer [53,54,55], there are no publications on *MXRA8* in breast cancer. Therefore, our work demonstrates for the first time that *MXRA8* is elevated in fast growing mammary tumors with a high metastatic potential. Our work also suggests that *MXRA8* is associated with breast cancer metastasis, as lung metastases expressed high levels of *MXRA8* and high *MXRA8* was associated with poor distant metastasis free survival in patients with basal-like breast cancer. Further support demonstrating that *MXRA8* plays a role in breast cancer stems from our work with murine mammary tumor cell lines. RJ423 cells, a murine mammary tumor cell line with claudin-low characteristics, express low levels of miR-200s and are highly tumorigenic and metastatic compared to RJ345 cells, which have some luminal characteristics [29,56,57,58]. A search of our RNA sequencing data revealed that RJ423 expressed significantly higher levels of *MXRA8* (logFC 6.9, adjusted pval 5.4 × 10^−66^) than RJ345 cells. Thus, in two different cell lines, *MXRA8* is elevated in aggressive, metastatic tumors expressing low miR-200s.

How *MXRA8* is regulated by miR-200c/141 will require additional work. While *MXRA8* has potential miR-200c or miR-141 binding sites, these sites were only predicted by miRWalk, and thus additional studies are required to determine whether *MXRA8* is a direct miR-200 target. In addition, miR-200c and miR-141 may also regulate the expression of genes like *MXRA8* indirectly through influencing the expression of transcription factors or altering the methylation status of DNA or histones. It is interesting to note that the top ENCODE and ChEA Consensus TFs from the ChIP-X pathway identified by Enrichr software was SUZ12. Our previously published transcriptome profiling of MDA-231c141 cells in 2D culture also found the top ENCODE and ChEA Consensus TFs from ChIP-X pathway to be SUZ12. SUZ12 is a member of the polycomb repressor complex 2 (PRC2) that mono-, di-, and tri-methylates histone H3 at lysine 27 (H3K27) [59,60]. Methylation of H3K27 is associated with decreased transcription. *MXRA8* is a predicted target of SUZ12 (Gene Set-SUZ12 (maayanlab.cloud)), and thus suppression of *MXRA8* expression in MDA-231c141 cells may be mediated by increased H3K27 methylation.

In summary, this study provides further evidence that elevated expression of the miR-200c/141 cluster is associated with reduced primary mammary tumor growth and metastasis. Moreover, this is the first study implicating *MXRA8* as a potential regulator of mammary tumor growth and metastasis.

## Figures and Tables

**Figure 1 genes-13-00691-f001:**
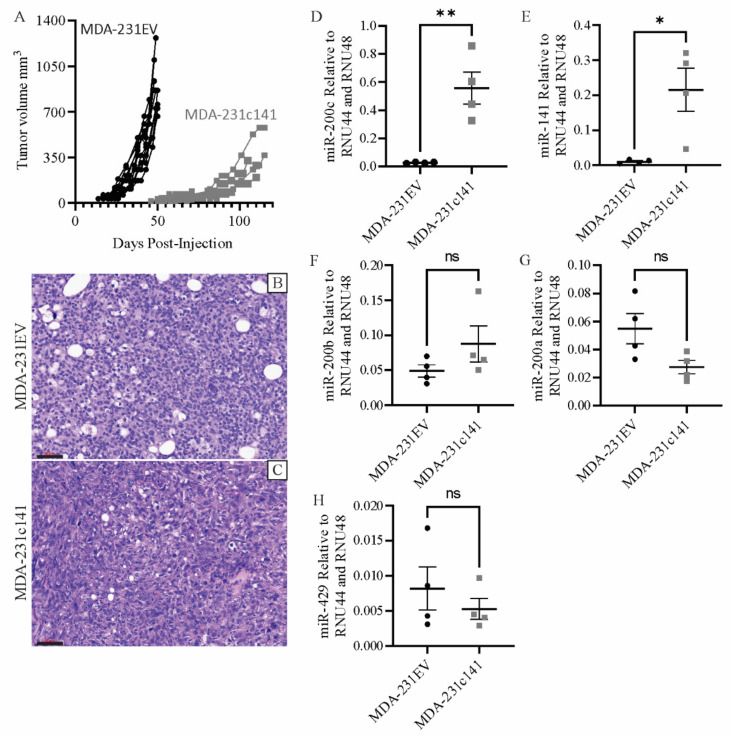
Mammary tumor growth curves for (**A**) MDA-MB-231 cells transfected with an empty vector (MDA-231EV) or the miR-200c/141 cluster (MDA-231c141). Hematoxylin and eosin stained sections of tumors from (**B**) MDA-231EV or (**C**) MDA-231c141 injections. Scale bars are 60 µm. Expression of (**D**) miR-200c, (**E**) miR-141, (**F**) miR-200b, (**G**) miR-200a, and (**H**) miR-429 in the tumors induced by MDA-231EV or MDA-231c141 injections. miR-200 values were normalized to RNU44 and RNU48. The individual values have been indicated along with the mean ± SEM. A Student’s *t*-test was performed and significance is indicated as * *p* < 0.05, ** *p* < 0.01, ns, not significant.

**Figure 2 genes-13-00691-f002:**
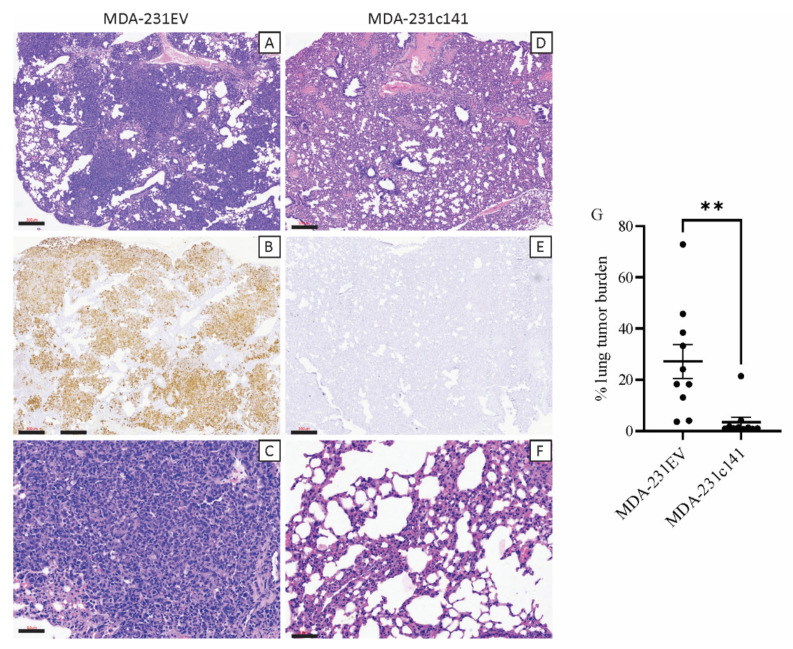
Lung sections from mice harbouring (**A**–**C**) MDA-231EV or (**D**–**F**) MDA-231c141 primary mammary tumors. Panels (**A**,**D**) show hematoxylin and eosin stained lung metastases at lower magnification while, panels (**C**,**F**) show representative lung sections at higher magnification. Panels (**B**,**E**) show staining of the lung metastases with a human, anti-vimentin antibody and the percentage of the lungs containing cells staining positive for human vimentin are quantified in (**G**). Scale bars of (**A**,**B**,**D**,**E**) are 300 µm while scale bars for (**C**,**F**) are 60 µm. (**G**) Quantification of lung tumor burden. To calculate lung tumor burden, the area of the lung with positive vimentin staining was expressed relative to the total lung area. The individual values have been indicated along with the mean ± SEM. A Student’s *t*-test was performed and significance is indicated as ** *p* < 0.01.

**Figure 3 genes-13-00691-f003:**
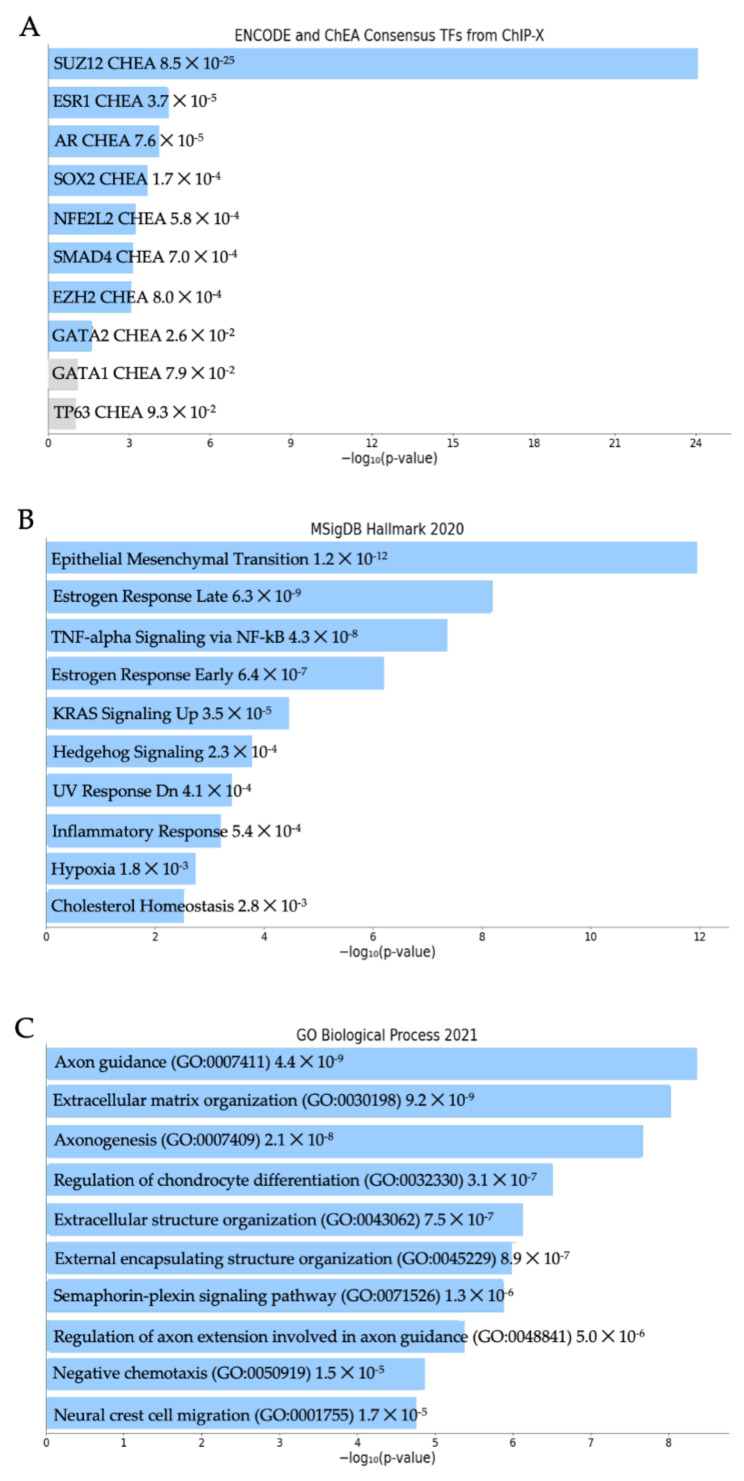
The top (**A**) ENCODE and ChEA Consensus TFs from ChIP-X, (**B**) MSigDB Hallmark 2020, and (**C**) GO Biological Process 2021 identified in Enrichr using the 2843 differentially expressed genes in MDA-231EV tumors compared to MDA-231c141 tumors.

**Figure 4 genes-13-00691-f004:**
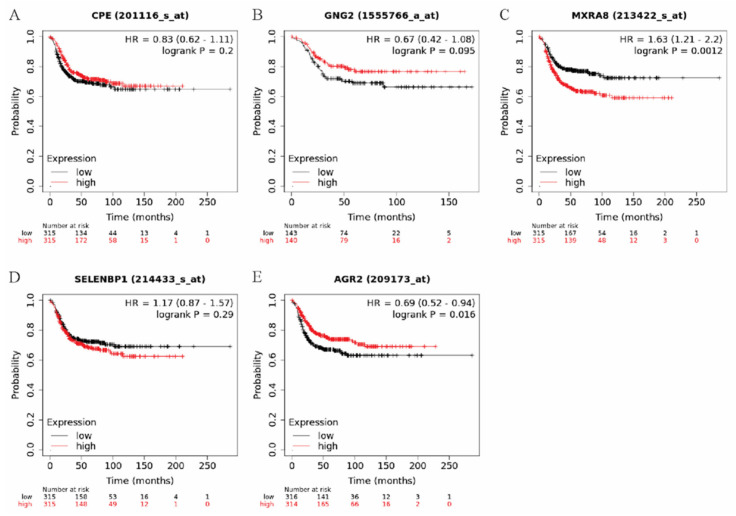
Distant metastasis free survival (DMFS) curves from Kaplan-Meier plotter using basal-like breast cancers and median cut-offs for (**A**) CPE, (**B**) GNG2, (**C**) MXRA8, (**D**) SELENBP1, and (**E**) AGR2. The red line indicates elevated gene expression while the black line indicates reduced gene expression. The hazard ratio (HR) and significance (logrank P) are indicated in the upper right hand corner of each graph.

**Figure 5 genes-13-00691-f005:**
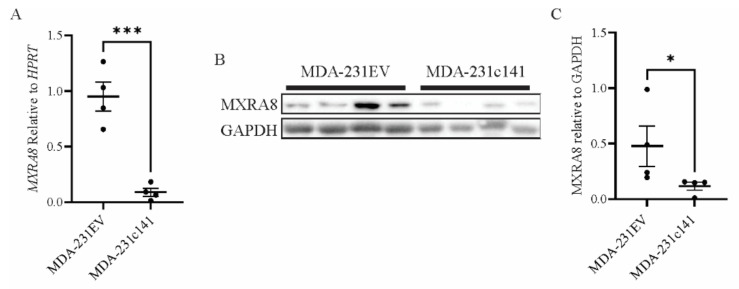
The expression of (**A**) MXRA8 mRNA levels as determined by qRT-PCR and (**B**,**C**) MXRA8 protein levels as determined by Western blotting. MXRA8 mRNA levels were normalized to HPRT, while MXRA8 protein levels were normalized using GAPDH. (**C**) Quantitation of the MXRA8 band intensities from the Western blot relative to the GAPDH band intensities. In both **A** and **C**, the individual values have been indicated along with the mean ± SEM. A Student’s *t*-test was performed and significance is indicated as * *p* < 0.05; *** *p* < 0.001.

**Figure 6 genes-13-00691-f006:**
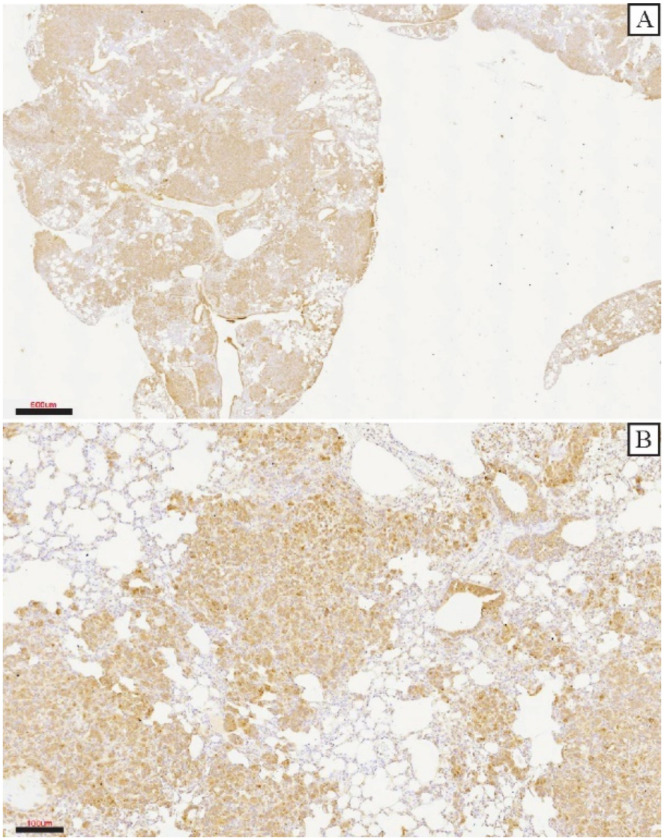
Immunohistochemistry at (**A**) low and (**B**) high magnification for MXRA8 in lung metastases from MDA-231EV primary tumors. Scales bars are 600 µm for (**A**) and 100 µm for (**B**).

**Table 1 genes-13-00691-t001:** Metastatic frequency of MDA-231EV and MDA-231c141 tumors.

	Lung	Pancreas/Spleen	Liver
MDA-231EV	10/10 (100%)	8/10 (80%)	6/10 (60%)
MDA-231c141	7/10 (70%)	2/10 (20%)	0/10 (0%)

**Table 2 genes-13-00691-t002:** Top five genes differentially expressed in MDA-231c141 vs. MDA-231EV tumors.

Transcript	logFC	FDR
*CPE*	−8.8	1.42 × 10^−223^
*GNG2*	−5.2	1.12 × 10^−210^
*MXRA8*	−7.9	2.50 × 10^−209^
*SELENBP1*	−7.4	4.88 × 10^−160^
*AGR2*	−10.0	3.44 × 10^−148^

## Data Availability

RNA sequencing has been uploaded to GEO under accession number GSE193479.

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
