# Peer review of "Elevated Expression of miR-200c/141 in MDA-MB-231 Cells Suppresses MXRA8 Levels and Impairs Breast Cancer Growth and Metastasis In Vivo"

_genes, 2022, doi:10.3390/genes13040691_

Round 1
Reviewer 1 Report
The authors investigated functional impact of restoring miR-200 expression in mesenchymal claudin-low breast cancer cell line MDA-MB-231 with extremely low levels of miR-200 family members. MDA-MB-231 cells were transfected with a control vector (MDA-231EV) or the miR-200c/141 cluster (MDA-231c141) and these cells were injected into mammary gland of mice. Using this mouse model of human breast cancer they found that MDA-231c141 significantly slowly produced tumors and spontaneous metastasis to the lung was significantly reduced compared to MDA-231EV. In MDA-231c141 tumors, MXRA8 gene expression was downregulated compared to MDA-231EV. Furthermore, the high levels of MXRA8 associated with reduced distant metastasis free survival and it was also detected in metastatic cells in the lung. They concluded that in mouse model, miR-200s inhibit growth and metastasis of claudin-low mammary tumor cells through downregulating MXRA8 expression.
The results of this study are interesting, but I have several questions and recommendation.
- Cell lines and culture condition: Are you sure that all details at your study are the same as in referenced study? The same question for other very short paragraphs in Material and Methods.
- The range of study is unknown. Number of injected mice, number of used transfectans….
- According to the revision of claudin-low breast cancer subtype (Fougner et al., Nature Communication, 2020), claudin–low cancers included 14.6% basal-like and 15.3% normal-like;therefore, the evaluation of DMFS and association of selected genes with breast cancer patient survival only in basal-like cancers is not adequate.
- The metastasized cells was detected only by vimentin expression. However, especially in MDA-231c141, the mesenchymal characters could be reverted into epithelial (your previous results and knowledge in targeting of EMT genes by miR-200s). Therefore, the identification of cancer cell spread could not be objective.
- Text at the rows 235-236 not correspond with Figure 6 legend.
- What could be a cause of discrepancy between your results and other authors (ref. 52, 53) (Discussion, rows 256-259)?
- What was the motivation for investigation of very specific and highly heterogeneous claudin-low subtype of breast cancer? Do you mean that your results could be applicable for other more aggressive subtypes or they could associate only with low claudin expression?
- There is surprising for me that the authors did not consider the effect of E-cadherin in evaluation of their results. After the overexpression of miR-200s in the cell line with mesenchymal character, the expression of EMT genes became be inhibited that allow expression of CDH1 gene with following impair of cell invasion.
- As you discussed, the overexpression of miR-200c/141 cluster was associated with downregulation of MXRA8 expression, but the direct in vitro interaction (targeting) between them was not showed. Therefore, formulate your interpretations more carefully.
Formal errors and recommendations:
- The authors forgot the text from instructions for authors in the paragraph Statistics (rows 134-148).
- Legends of figures and tables are written in incorrect font.
- References 29 and 37 are accepted and published version of the same paper.
- Enrich (software?), GEO (database?), ImageJ (software?)
This version of manuscript need to be re-evaluate according to above–mentioned recommendations. I recommend a major revision.
Author Response
I would like to thank the reviewer for their comments. Each comment and how I have addressed each comment in the revised manuscript is listed below.
- Cell lines and culture condition: Are you sure that all details at your study are the same as in referenced study? The same question for other very short paragraphs in Material and Methods
I have expanded the material and methods to include more details about the cell lines and this information is found in lines 69-74. Yes, the cell lines are the same as those described in Simpson et al. Cancer Cell Int 2021.
- The range of study is unknown. Number of injected mice, number of used transfectants
I have included more details on the injection of tumor cells into the NCG mice in the methods and this information is found in lines 85-88. The stable selection of MDA-231EV and MDA-231c141 cells was performed on bulk cell cultures and thus the number of transfectants was not reported. The fact that tumor growth curves were consistent despite injections occurring several weeks/months apart demonstrate that the phenotype was maintained in the selected cells.
- According to the revision of claudin-low breast cancer subtype (Fougner et al., Nature Communication, 2020), claudin–low cancers included 14.6% basal-like and 15.3% normal-like;therefore, the evaluation of DMFS and association of selected genes with breast cancer patient survival only in basal-like cancers is not adequate.
The purpose of this figure was to demonstrate the clinical relevance of MXRA8 in breast cancer metastasis. We chose basal-like breast cancer as basal-like breast cancer and claudin-low breast cancer share a number of features. In the discussion we use this information to support our hypothesis that MXRA8 may influence breast cancer metastasis. We do not specifically state that MXRA8 is important for metastasis in patients with claudin-low breast cancer.
- The metastasized cells was detected only by vimentin expression. However, especially in MDA-231c141, the mesenchymal characters could be reverted into epithelial (your previous results and knowledge in targeting of EMT genes by miR-200s). Therefore, the identification of cancer cell spread could not be objective.
The vimentin-stained slides essentially mirrored the lung metastases observed in the H&E sections and we confirmed this in all the lung sections. We did not observe any lung metastases that did not stain with the vimentin antibody in either the MDA-231EV or MDA-231c141 lung metastases. The H&E sections are difficult to quantify since normal and tumor cells stain the same. In addition, the RNA sequencing of the MDA-231EV and MDA-231c141 tumors did not reveal any significant differences in vimentin expression. The average vimentin expression for MDA-231EV tumors was 4219 and the average vimentin expression for the MDA-231c141 tumors was 3688 (using transcripts per million counts). Thus, the cells in both tumors express high levels of vimentin.
- Text at the rows 235-236 not correspond with Figure 6 legend.
In the original manuscript, lines 235-236 were describing the MXRA8 staining in the murine lung tissue which is shown in Figure 6. The lung metastases stain brown while adjacent normal lung tissue does not stain for MXRA8. I have added the word adjacent in the revised manuscript, line 259 to help clarify this.
- What could be a cause of discrepancy between your results and other authors (ref. 52, 53) (Discussion, rows 256-259)?
The main difference between the studies referenced in 51 and 52 of the revised manuscript is that these studies utilized tail vein injection of tumor cells while our study evaluated spontaneous metastasis to the lungs from a primary mammary tumor. The discussion indicates that these two studies utilized tail vein injections which do not mimic all the steps required for spontaneous metastasis.
- What was the motivation for investigation of very specific and highly heterogeneous claudin-low subtype of breast cancer? Do you mean that your results could be applicable for other more aggressive subtypes or they could associate only with low claudin expression?
Our work on claudin-low breast was driven from our mouse models of mammary tumorigenesis. We have a murine mammary tumor cell line that has claudin-low characteristics. When miR-200s were overexpressed in this murine claudin-low mammary tumor cell line, mammary tumor growth and metastasis were inhibited (Exp Cell Res 369:17-26, 2018). We wanted to investigate whether miR-200s had similar effects in human breast cancer and chose a claudin-low cell line to be consistent with our murine studies. We are currently looking at the impact of increased miR-200 expression in human HER2+ breast cancer cells as well as the role of MXRA8 in metastasis of HER2+ breast cancer, but we do not have this data yet.
- There is surprising for me that the authors did not consider the effect of E-cadherin in evaluation of their results. After the overexpression of miR-200s in the cell line with mesenchymal character, the expression of EMT genes became be inhibited that allow expression of CDH1 gene with following impair of cell invasion.
We evaluated E-cadherin (CDH1) levels in the MDA-231EV and MDA-231c141 cells (in vitro) using RNA-seq and qRT-PCR and tumors using RNA-seq data and found no significant change in CDH1 levels. Therefore, we did not investigate E-cadherin further.
- As you discussed, the overexpression of miR-200c/141 cluster was associated with downregulation of MXRA8 expression, but the direct in vitro interaction (targeting) between them was not showed. Therefore, formulate your interpretations more carefully.
I believe we have been careful. We mention three possible ways (directly or indirectly through transcription factors or epigenetics) that miR-200s may influence MXRA8 expression in the discussion. We never claim that miR-200s directly regulate MXRA8 but describe what we found and that is one software (miRwalk), indicated that MXRA8 has a potential miR-200c/141 binding sites. We are currently trying to determine how miR-200s may regulate MXRA8 levels by initially looking at histone methylation in the MXRA8 promoter but we do not have this data yet.
- The authors forgot the text from instructions for authors in the paragraph Statistics (rows 134-148).
This text has now been deleted
- Legends of figures and tables are written in incorrect font.
Figure legends now have the same font as the manuscript. The figures were created in CorelDraw and the font is Palatino Linotype 10pt (same as the manuscript). However, the font in the figures legends still looks slightly different than the font in the text of the manuscript. I’m not sure how to fix this.
- References 29 and 37 are accepted and published version of the same paper.
These references have been fixed
- Enrich (software?), GEO (database?), ImageJ (software?)
We have added this information in the revised manuscript.
Reviewer 2 Report
The manuscript of Simpson et al. describes functional impact of restoring miR-200 expression in a human claudin-low breast cancer cell line MDA-MB-231.
The manuscript is potentially interesting for experts in the field. However, I have several comments / recommendations / questions:
- The authors should explain better the intent of their work.
- The section Materials and Methods must be improved.
- How many mice were used? Moreover, in the section Animals and Ethics, only from the second paragraph it became clear that we were talking about mice.
- Sections RNA Extraction and RNA Sequencing, Real-Time PCR, Western blotting should be written in more detail.
- In the section Kaplan-Meier Plotter in sentence “Patients were split by the median value” word “Patients” better replace with “animals” or “mice”
- In the section “Statistics” contains text from the manuscript template
- RNU44 and RNU48 is a long obsolete control and is not used for normalization of miRNA expression TODAY. Spike-control is absent!
- etc…
3. General impression: the article is written extremely carelessly(
- “In vivo” usually written in italics.
- the scale bars in Figure 1B,C are almost invisible.
- not enough captions for figures 1D-H, 2G, 5 A,C
- the manuscript contains 5/19 half-blank pages
- some references from somewhere copied and not updated. For example, ref 61:
Martin, C.M., RA. Polycomb repressor complex 2 function in breast cancer. International journal of oncology 2020, (in press).
Author Response
I would like to thank the reviewer for their comments. Each comment and how I have addressed each comment in the revised manuscript is listed below.
- The authors should explain better the intent of their work.
A statement has been added to the introduction indicating the goal of this study, lines 59-61
- The section Materials and Methods must be improved.
Additional details have been added to the material and methods
- How many mice were used? Moreover, in the section Animals and Ethics, only from the second paragraph it became clear that we were talking about mice.
We have stated mice were used earlier under the heading “Animals and ethics” line 78 and included the number of mice used, lines 84-86
- Sections RNA Extraction and RNA Sequencing, Real-Time PCR, Western blotting should be written in more detail.
I have added a few additional details but these methods have previously been published. I can repeat the methods in the revised manuscript but most journals want to limit replication of published protocols.
- In the section Kaplan-Meier Plotter in sentence “Patients were split by the median value” word “Patients” better replace with “animals” or “mice”
The Kaplan-Meier Plotter used data from human patients. Thus, patients is the most appropriate term.
- In the section “Statistics” contains text from the manuscript template
This has been removed.
- RNU44 and RNU48 is a long obsolete control and is not used for normalization of miRNA expression TODAY. Spike-control is absent!
Actually, a paper published in Eur J Clin Nutr 2021 concluded that RNU44 “RNU44 as suitable reference miRNAs for placental samples…” and a paper published in Oncol Lett 2019 indicated that RNU44 and RNU48 were suitable for normalization. Similarly, a paper published in Exp Oncol 2021 used RNU48 as a reference miRNA. There are also other examples of researchers using RNU44 and/or RNU48 for miRNA normalization.
In addition, RNU44 and RNU48 have very stable expression in the MDA-231EV and MDA-231c141 cells. For example a run of 3 distinct MDA-231EV samples had RNU44 Cq values that ranged from 19.3-20.2 and RNU48 Cq values that ranged from 20.0-20.1 while 3 distinct MDA-231c141 samples had RNU44 and RNU48 Cq values that ranged from 18.6-20.7 and 19.1-20.1 respectively.
- “In vivo” usually written in italics.
This is often journal dependent as some journals use italics for in vitro and in vivo while other do not. I have changed in vivo and in vitro to italics in the revised manuscript
- the scale bars in Figure 1B,C are almost invisible.
The scale bars have been modified to make them more visible. I was unable to modify the µm values as this is generated by the software, however, the µm values are listed in the figure legends.
- not enough captions for figures 1D-H, 2G, 5 A,C
I have added additional details to these figure legends.
- the manuscript contains 5/19 half-blank pages
I did not realize that the figures should be sized for the final paper form. I left the figures larger so the reviewers could see the data more clearly. I have altered the size of the figures in the revised manuscript
- some references from somewhere copied and not updated. For example, ref 61:
These references have been fixed.
Round 2
Reviewer 1 Report
I accept all explanations and revisions in the manuscript except following:
- In consistency with your explanation, I recommend to insert into the paragraph Histology and immunohistochemistry the sentence that all MDA-231EV and MDA-231c141 cells in lung were identified by high levels of vimentin.
- Specify A and B in Figure 6. Legend.
- Subsequently, the answer on my question is that you do not consider some associations of your results and claudin-low phenotype. However, it is known that claudins have several roles in cancer metastasis.
Author Response
- In consistency with your explanation, I recommend to insert into the paragraph Histology and immunohistochemistry the sentence that all MDA-231EV and MDA-231c141 cells in lung were identified by high levels of vimentin
I have included a statement indicating that metastatic tumor cells stained positive for vimentin; lines 107-109
- Specify A and B in Figure 6. Legend.
I have modified the figure legend of Figure 6 to indicate (A) displays a low magnification image, while (B) displays a high magnification image of MXRA8 staining in lung metastases.
- Subsequently, the answer on my question is that you do not consider some associations of your results and claudin-low phenotype. However, it is known that claudins have several roles in cancer metastasis.
I apologize but I am not sure what is meant by this comment as I don’t know which original concern you are referring to. There was a comment about the role of E-cadherin in invasion in the original list of concerns however we did not find any difference in E-cadherin expression between MDA-231c141 and MDA-231EV tumors. Similarly, there was no significant change in the levels of claudins 3, 4 or 7 in the MDA-231c141 tumor compared to the MDA-231EV tumors. Now, expression levels cannot infer function so I cannot conclude that E-cadherin or claudins do not play a role. It’s just given the lack of change in expression, these gene/proteins were not evaluated.
Reviewer 2 Report
The manuscript has been revised, but the number of comments has decreased slightly. In particular,
- How many cells and tissues (g) were used for RNA isolation? How much RNA was taken for sequencing?
- Sections Real-Time PCR and Western blotting remain as short and obscure as ever. Of course, it is not necessary to rewrite the methodology, but the fundamental points, for example, how much material was taken and how much RNA was isolated - this is important for understanding the possibility of carrying out the experiment.
- “Patients” better replace with “animals” or “mice”
- There is no evidence of the purity and quality of the isolated RNA, which casts doubt on the results. Please add these data (for example OD260/280 and capillary electrophoresis).
- In the captions to figures 1D-H, 2G, 5 A,C it remains unclear what the scope and line reflect. 25%-75%? Median/mean?
- Really, in the manuscript “Placental expression of RNU44, RNU48 and miR-16-5p: stability and relations with fetoplacental growth” (Eur J Clin Nutr. 2021 Sep 10. doi: 10.1038/s41430-021-01003-3) the authors concluded that miR-16-5p and RNU44 are suitable reference miRNAs for placental samples. However, it should be noted that in current manuscript authors didn’t work with placenta, they estimated miRNA level in breast cancer cells. The main disadvantage of RNU44 and RNU48 is the fact that they are significantly longer (>100 bp) than miRNAs, leading to differences in isolation efficiency and RT-PCR. Moreover, quantification of RNU6B, RNU44, and RNU48 in healthy females and breast cancer patients showed that snRNAs were the least stable reference genes, as derived from their microarray dataset of 380 miRNAs (McDermott AM, Miller N, Wall D, Martyn LM, Ball G, Sweeney KJ et al (2014) Identification and validation of oncologic miRNA biomarkers for luminal A-like breast cancer. PLoS One 9(1):e87032). Finally, Gee et al. showed that snoRNAs had highly variable expression levels in breast cancer patients and were associated with their clinicopathological factors. For example, the amounts of RNU48 were negatively correlated with tumor grade, those of RNU48 and RNU43 were inversely correlated with proliferation score, and lower levels of RNU44 were an adverse prognostic factor for overall survival (Gee H, Buffa F, Camps C, Ramachandran A, Leek R, Taylor M et al (2011) The small-nucleolar RNAs commonly used for microRNA normalisation correlate with tumour pathology and prognosis. Br J Cancer 104(7):1168), suggesting that these snoRNAs have tumor suppressive features. To ensure that miRNA quantification is not affected by the technical variability that may be introduced at different analysis steps, synthetic, non-human spike-in miRNAs should be used. The Caenorhabditis elegans miRNA cel-miR-39 as an exogenous control is frequently used for data normalization (Armand-Labit V, Pradines A (2017) Circulating cell-free microRNAs as clinical cancer biomarkers. Biomol Concepts 8(2):61–81; Stevic I, Müller V, Weber K, Fasching PA, Karn T, Marmé F et al (2018) Specific microRNA signatures in exosomes of triple-negative and HER2-positive breast cancer patients undergoing neoadjuvant therapy within the GeparSixto trial. BMC Med 16(1):179) have also been used. The addition of this exogenous miRNA to the samples before reverse transcription of RNA documents the uniform handling of the samples and RT-PCR efficiencies. This spike-in method can additionally eliminate deviations of the experimental process and, thus, provides more reliable results. However, the use of such a spike-in control does not correct for deviations in sampling or quality of the samples.
THEREFORE, DATA NORMALIZATION SHOULD ALWAYS BE CARRIED OUT BY A COMBINATION OF AN ENDOGENOUS AND AN EXOGENOUS CONTROL MIRNA, TO WARRANT THAT SUCH DIFFERENCES IN MIRNA DETECTION MAY BE COMPENSATED.
Author Response
1. How many cells and tissues (g) were used for RNA isolation? How much RNA was taken for sequencing?
RNA was isolated from four independent MDA-231EV and four independent MDA-231c141 tumors. All samples had a 260/280 ratio > 1.95 (Nanodrop) and ~2ug of RNA was sent to Genome Quebec. Genome Quebec analyzed the RNA on an Agilent Bioanalyzer 2100 (Agilent, Santa Clara, CA) and the samples had an RIN of 9.5 or greater. Libraries were generated from 250ng of total RNA which included mRNA enrichment using the NEBNext Poly(A) Magnetic Isolation Module (New England BioLabs, Whitby, ON, Canada), cDNA synthesis using NEBNext RNA First Strand Synthesis and NEBNext Ultra Directional RNA Second Strand Synthesis Modules (New England BioLabs, Whitby, ON, Canada). The libraries were normalized, pooled and denatured in 0.05N NaOH and neutralized using HT1 buffer. The pool was loaded at 225pM on an Illumina NovaSeq S4 lane using Xp protocol as per the manufacturer’s recommendations. The run was performed for 2x100 cycles (paired-end mode). A phiX library was used as a control and mixed with libraries at 1% level. Base calling was performed with RTA v3. Program bcl2fastq2 v2.20 was then used to demultiplex samples and generate fastq reads.
This information has now been added to the RNA Extraction and RNA Sequencing Section of the methods lines 116-129.
2. Sections Real-Time PCR and Western blotting remain as short and obscure as ever. Of course, it is not necessary to rewrite the methodology, but the fundamental points, for example, how much material was taken and how much RNA was isolated - this is important for understanding the possibility of carrying out the experiment.
I have expanded the details in these two sections, lines140-167 and lines 170-182.
3. “Patients” better replace with “animals” or “mice”
I removed the term patients from our discussion related to the role miR-200c/141 in metastasis as this worked was performed in mouse models (line 372). Any data relating to KM plotter was performed using patient breast cancer samples and thus “patients” is the most appropriate term here. Using “animals” or “mice” would be misleading.
4. There is no evidence of the purity and quality of the isolated RNA, which casts doubt on the results. Please add these data (for example OD260/280 and capillary electrophoresis).
This information has now been included in the methods (lines 117-119) when describing the RNA Extraction and RNA sequencing. The same RNA that was used for sequencing was also used for Taqman and standard RT-PCR
5. In the captions to figures 1D-H, 2G, 5 A,C it remains unclear what the scope and line reflect. 25%-75%? Median/mean?
The figure legends have been modified to indicate the graphs represent the mean ± SEM.
6. RNU44 and RNU48 as housekeeping genes
I looked at the paper by McDermott et al, PLoS One 9:387032, 2014 and I cannot find the data in this paper indicating that RNU44 and RNU48 were the least stable reference genes. The microarray dataset is available in GEO so maybe the reviewer re-analyzed the data, I did not attempt to re-analyze this data. My concern with this paper, as it relates to our data, is that the McDermott paper analyzed microRNA levels in blood samples not solid tumors (except for a few miRNAs). We know miRNA expression analysis in blood can be tricky as collection of these samples require additional care to prevent hemolysis and the introduction of miRNAs from red blood cells into the sample. Also, the levels of miRNAs in the blood are quite low so normalization is an extremely important consideration.
What is interesting is that the McDermott paper indicates that they used miR-16 as the housekeeping gene and then referenced two papers, one of which was Davoren et al BMC Mol Biol 9:76, 2008. In the Davoren paper, GeNorm was used for to identify potential housekeeping sequences in human primary breast cancers and RNU48 stability was just slightly higher (0.401) than miR-16 (0.379) indicating that RNU48 is almost as good as miR-16 when normalizing miRNA levels in solid breast cancers. Since the Davoren study was performed in mammary tissue it is more relevant to our study than the McDermott paper that was evaluating blood samples from cancer patients.
What I also found interesting is that neither the McDermott nor Davoren paper used spike-in controls.
The part I really don’t understand is why the reviewer is putting so much emphasis on this point. We only used miRNA quantification in this study to confirm miR-200c and miR-141 expression were maintained at elevated levels in the cells following injection and tumor formation. As indicated to the reviewer in our first response the Cq values for RNU44 and RNU48 were very similar for the MDA-231EV and MDA-231c141 tumors. The average Cq levels for RNU44 in MDA-231EV tumors was 20.0 and in MDA-231c141 tumor it was 19.9. Similarly for RNU48, the average Cq values in MDA-231EV tumors was 19.7 and in MDA-231c141 tumors it was 20.4. Thus, both RNU44 and RNU48 are highly stable (less than 1 cycle different) in MDA-231 cells and tumors. Does this mean that RNU44 and RNU48 will be suitable normalizers in all breast cancer studies…no, probably not.
When looking at miR-200c levels, the average Cq value for MDA-231EV tumors was 25.3 and for MDA-231c141 tumors it was 21.1 (4 cycles different). Similarly for miR-141, the average Cq values for the MDA-231EV tumors was 26.6 and for the MDA-231c141 tumors it was 22.7 (again, almost 4 cycles different). Also, if the increase in miR-200c and miR-141 expression was due to a decrease in RNU44 and RNU48 expression, would one not expect miR-200b, miR-200a and miR-429 to also be elevated in the MDA-231c141 tumors compared to the MDA-231EV tumors since RNU44 and RNU48 were also used to normalize these miRNAs?
In addition, when we first created the MDA-231EV and MDA-231c141 cells, miRNA-sequencing was performed. This sequencing showed that miR-200c (logFC 7.1, pval 0) and miR-141 (logFC 5.3, pval 1.5x10-58) were significantly elevated in the MDA-231c141 cells compared to the MDA-231EV cells. miR-200c was the top differentially expressed miRNA based on pval in MDA-231c141 cells compared to MDA-231EV cells. This data was confirmed using Taq-Man RT-PCR with RNU44 and RNU48 as normalizers and this work is published in Cancer Cell International 21:89, 2021. Again RNU44 and RNU48 were very similar in the MDA-231EV and MDA-231c141 cells.
Given all of this data, we are confident in our conclusion that high levels of miR-200c and miR-141 are maintained in the MDA-231c141 cells following their injection and mammary tumor formation.
Round 3
Reviewer 2 Report
Everything is fine/
Sorry for the delay - was on a business trip.